# Cataract-Associated New Mutants S175G/H181Q of βΒ2-Crystallin and P24S/S31G of γD-Crystallin Are Involved in Protein Aggregation by Structural Changes

**DOI:** 10.3390/ijms21186504

**Published:** 2020-09-05

**Authors:** In-Kang Song, Seungjin Na, Eunok Paek, Kong-Joo Lee

**Affiliations:** 1Department of Pharmacy, College of Pharmacy and Graduate School of Pharmaceutical Sciences, Ewha Womans University, Seoul 03760, Korea; fluss1234@ewha.ac.kr; 2Department of Computer Science, Hanyang University, Seoul 04763, Korea; sna@hanyang.ac.kr

**Keywords:** βΒ2-crystallin, γD-crystallin, cataract-associated mutants, proteomics dataset, hydrogen-deuterium exchange-MS, structural change, stability change, post translational modification, protein aggregation

## Abstract

β/γ-Crystallins, the main structural protein in human lenses, have highly stable structure for keeping the lens transparent. Their mutations have been linked to cataracts. In this study, we identified 10 new mutations of β/γ-crystallins in lens proteomic dataset of cataract patients using bioinformatics tools. Of these, two double mutants, S175G/H181Q of βΒ2-crystallin and P24S/S31G of γD-crystallin, were found mutations occurred in the largest loop linking the distant β-sheets in the Greek key motif. We selected these double mutants for identifying the properties of these mutations, employing biochemical assay, the identification of protein modifications with nanoUPLC-ESI-TOF tandem MS and examining their structural dynamics with hydrogen/deuterium exchange-mass spectrometry (HDX-MS). We found that both double mutations decrease protein stability and induce the aggregation of β/γ-crystallin, possibly causing cataracts. This finding suggests that both the double mutants can serve as biomarkers of cataracts.

## 1. Introduction

Crystallins are a group of structural proteins located in the eye lens that function to focus light onto the retina. Cataracts, the leading cause of blindness worldwide [1], develop due to the misfolding and aggregation of crystallins [2,3]. The mammalian lens consists of 90% of soluble protein crystallins [4], which have a lifespan of more than 70 years [5]. Three lens crystallins (α-, β-, and γ-crystallins) are classified into two families, α-crystallins and βγ-crystallins. α-Crystallins function as molecular chaperones by preventing the aggregation of βγ-crystallins and maintaining their solubilities [6] with similar way to small heat shock proteins, and βγ-crystallins play key roles as structural proteins for maintaining the structural stability of lens [7]. βγ-Crystallins are composed primarily of two double-Greek key motifs [8,9], which confer stability and longevity to the parent molecules. Crystallins are encoded in the CRY genes, and to date, more than 100 mutations have been identified in 12 crystallin genes isolated from more than 100 families. They account for 33–37% of autosomal dominant cataracts [10]. In the mutations of CRYAA and CRYAB encoding the α-crystallins, the former is associated with nuclear and lamellar-type opacities, and the latter to myopathy. Mutations in the genes for β-crystallins (CRYBB1, BB2, BB3, BA1, BA2, and BA4) and γ-crystallins (CRYGB, GC, GD, and GS) interfere the folding of structurally conserved Greek-key motifs or change the surface properties of proteins. Mutations reducing the solubility causing the aggregation of βγ-crystallins are associated with congenital or juvenile cataract [11].

βB2-crystallin, the least modified and the most soluble β-crystallin, is the predominant β-crystallin in human eye lens [12]. It exists as homo- and hetero-dimers and oligomers with other β-crystallins [13]. There are two known cataract-causing mutants: A2V has a poor thermal stability due to its poor ability to make tetramers at high concentrations [14], and Q155X has a structure, that makes it less stable [15]. γD-crystallin is highly stable and is not denatured by conventional denaturing agents and heat [16]. The *N*-terminal domain of γD-crystallin is less stable than the C-terminal domain [17]. The mutants identified in the *N*-terminal domain of γD-crystallin are as follows: R15C mutation causes protein aggregation as result of the formation of disulfide-linked oligomers [18]. P24S, P24T, R37S, and R59H mutations decrease the protein solubility and crystallize easily [8,19,20,21]. W43R mutation increases the proteolytic susceptibility and facilitates to aggregate [22].

In this study, we found two new nuclear cataract-associated mutants in βB2- and γD-crystallins by an analysis of the proteomics dataset of cataract patients employing bioinformatics tools followed by a study of the effects of these mutations, S175G/H181Q on βB2-crystallin and P24S/S31G on γD-crystallin. Both two double mutations are located in the largest loops connecting two distant β-sheets in the Greek key motif. We examined the structural changes employing spectroscopic and mass spectrometric analysis. We found significant structural changes of βB2-crystallin S175G/H181Q mutant and demonstrated perturbation of the C-terminal Greek key 4 motifs by studying protein structural dynamics via HDX-MS [23]. We also found structural changes of γD-crystallin P24S/S31G mutant, but these are only in acidic condition, because of the high stability of γD-crystallin in normal condition. We found that increase of cataract-related modifications including oxidation occur only in this mutant. These studies reveal that two new mutations found in this study are located in the largest loop between distant β-sheets in Greek key motif. We believe they promote cataract development due to the structural changes of Greek key motif. They could serve as biomarkers of cataract development.

## 2. Results

### 2.1. Selection of Two New Nuclear Cataract-Specific Mutations by Bioinformatics Analysis

Five normal human lenses (3-day and 2-, 18-, 35-, and 70-year old) and two nuclear cataract lenses (70- and 93-year old) were obtained from previous studies, where age-related changes in human crystallins were analyzed [24,25]. To define cataract-specific amino acid mutations in crystallin proteins, we retained peptides that were identified with amino acid mutations from 14 crystallin proteins. The mutations were consistently observed in both 70- and 93-year cataract lens samples but were not observed in other normal lens samples. Mutations were accepted only when there were no possible chemical modifications matching the delta masses. The resulting ten peptides are shown with the amino acid mutations in Table 1.

To determine the characteristics of each mutation, we examined the location of each mutation in the structure (Appendix A). Two of the mutants are in α-crystallin and eight are in β/γ-crystallin. αB-Crystallin, one of α-crystallins having a HSP domain, has a crystal structure in which only some structures have been identified (PDB ID code 2Y1Y) [26]. In αB-crystallin, the location of Phe61 is not yet known, but that of Thr132 is in the helix between immediately adjacent β-sheets (Appendix A). Unlike α-crystallin, β/γ-crystallins have four Greek key motifs (Appendix A). The Greek key motif is known to be evolutionarily well conserved super-secondary protein structural fold that affords structural compactness and high intrinsic stability against various stress including ROS and aging [17,27]. The Pro185 of βA4-crystallin is located in a loop near the C-terminal of the Greek key motif 4, where the structure is not clear (PDB ID code 3LWK) (Appendix A). Ser227 of βB1 (PDB ID code 1OKI) and Ile124 of βB2-crystallin (PDB ID code 1YTQ) are located at the end of β-sheet of Greek key motif 3 and 4 (Appendix A). In βB3-crystallin, Ser71 and 125 are at the end of β-sheet of 2 and 3 motif, and Serine 187 is in the loop at end of Greek key 4 motif (PDB ID code 3QK3) (Appendix A). Intriguingly, two double mutations found in βB2- and γD-crystallin are very close in sequence and were identified in one tryptic digested peptide. Both Ser175 and His181 of βB2-crystallin and Pro24 and Ser31 of γD-crystallin are located in the largest loop connecting to distant β-sheets in the Greek key motif (Appendix A). Although large loops do not form disulfide or coordinate with metals in βB2 and γD-crystallins, these large loops are important to maintain stability of the Greek key motif [28]. We selected and validated these two double mutations employing recombinant proteins of these mutants.

### 2.2. Structural Characteristics of βB2-Crystallin Double Mutants

In order to investigate the structural changes caused by these mutations of βB2-crystallin, we examined the intrinsic fluorescence changes of Trp and Tyr of recombinant wild-type (WT) and S175G/H181Q βB2-crystallin. The intrinsic fluorescence maximum wavelength and the intensity of aromatic amino acids (Trp and Tyr) were measured, because intrinsic fluorescence of Trp and Tyr varies depending on the structural environment of these residues [29]. As shown in Figure 1, the changes of fluorescence signals were induced by the double mutation S175G/H181Q βB2-crystallin. The maximum emission wavelengths of intrinsic Trp fluorescence (λmax), the strongest intrinsic fluorescence, were 334 nm for WT, an about 5 nm red-shift by the mutation could be observed for the fluorescence (Figure 1a). The maximum emission wavelengths of intrinsic Trp & Tyr fluorescence were 340 nm for both WT and mutant without red shift (Figure 1b). The intensities of intrinsic Trp and Trp & Tyr fluorescence of the mutant were 1.1~1.3 times higher than those of WT. This change is meaningful because these changes are sum of different 9 Tyr and 4 Trp residues. These spectroscopic results indicate that the mutations affect the conformational state and the microenvironments of the Trp and Tyr fluorophores. βB2-crystallins are known to form homodimers and heteromers with other β-crystallins [13,30] which are required to increase the solubility. Since higher oligomeric fraction of βB2-crystallin increases the solubility of the other β-crystallins [31,32], we examined the dimeric states of WT and S175G/H181Q mutant using size exclusion chromatography (SEC). As shown in Figure 1c, dimeric fraction of WT βB2-crystallin is higher than that of mutant form, although all β-crystallins migrate on SEC as dimers in rapid equilibrium with monomer form. This shows the possibility that S175G/H181Q mutation destabilizes dimer formation and makes the mutant protein less soluble than WT one.

### 2.3. Effects of Mutations on βB2-Crystallin Stability and Folding

To investigate the stability differences between WT and S175G/H181Q mutant βB2-crystallins, we assessed red shift of intrinsic fluorescence changes by chemical denaturant GdnHCl in a dose dependent manner. As shown in Figure 2a, mutant at 0.5~0.6 M GdnHCl is more readily denatured than WT at 0.6~0.8 M. To validate these results, we performed ANS binding assay in response to various concentrations of GdnHCl, since significant increase of ANS fluorescence can be observed by increasing hydrophobic surface in denatured and aggregated form of protein [33]. WT βB2-crystallin did not affect the hydrophobic exposure during GdnHCl-induced unfolding, while S175G/H181Q mutant significantly increased the hydrophobic exposure at low 0.6~0.8 M GdnHCl concentrations (Figure 2b). This increase of ANS fluorescence is caused by the accumulation of denatured form having large hydrophobic exposure or the appearance of aggregates.

To evaluate the long-term stability, the protein with a concentration of 5 mg/mL were incubated at around physiological temperature (37 °C) for 36 h and measured turbidity (A_400_). As shown in Figure 2c, WT protein maintained transparent after 18 h, while the turbidity of mutant increase continuously till 36 h incubation. That is, the mutant protein is susceptible to aggregation induced long-term incubation. This result also confirmed that, unlike WT, when mutants were overexpressed in mammalian cells, the mutants were aggregated in a small puncta form (Appendix A). These results indicate that S175G/H181Q mutation of βB2-crystallins reduced protein stability by affecting the protein folding and increased the tendency to be aggregated.

### 2.4. Identification of Structural Changes in Mutant βB2-Crystallin Employing Hydrogen Deuterium Exchange Mass Spectrometry (HDX-MS)

To determine the structural changes induced by mutation, we predicted mutant structure employing protein modeling analysis using the Phyre2 software [34]. As shown in Figure 3a, mutation induced significant structural changes in the C-terminus, but not in the *N*-terminus. Based on these predicted results, we assumed that mutation caused the clumping of the C-terminal Greek key motif, which is important for the stability of βB2-crystallin [35], causes protein aggregation. To confirm this possibility, we employed hydrogen-deuterium exchange-mass spectrometry (HDX-MS), which measures the differential deuterium incorporation in proteins and peptides between WT and S175G/H181Q mutant βB2-crystallins and is a powerful tool to elucidate structural changes [36]. Differential deuterium incorporations of identified peptides of WT and mutant in a time dependent manner are listed in Appendix A. Combined stitching the H/D exchange ratios of each peptic peptide showed the diagram of whole protein shown in Figure 3b. Peptide MS coverage was 98%. Significant conformational changes were observed in the Greek key motif 4 of βB2-crystallin (Figure 3c). More deuterium incorporation in mutant occurred in the peptide containing the portion between the Greek key motifs 3 and 4 (144–151) and the largest loop of the Greek key motif 4 (166–186). These results suggest the Greek key motif 4 with apparent roles in βB2-crystallin structure and assembly [35] is exposed to the protein surface by mutation and is more flexible than WT. This destabilizes the Greek key motif 4 structure. Meanwhile, less deuterium exchange was observed in peptide located in the Greek key motif 2 (62–72). This indicates that the Greek key motif 2 is more shielded by mutation. Here, by confirming the change not only in the substituted site, but also in other sites, it can be assumed that the observed HDX change is the result of solvent exposure due to structural flexibility and stability, and not due to the effect of amino acid substitution on intrinsic exchange rate. These results indicate that S175G/H181Q mutations of βB2-crystallin cause the conformational changes in the Greek key motif 4 which play a role in maintaining stable structure of βB2-crystallin.

### 2.5. Effect of the Mutations on γD-Crystallin Structure Revealed by Biophysical Experiments

Another new γD-crystallin double mutant (P24S/S31G) was identified in cataract lenses (Table 1). Since single P24S mutant has already been studied in the cataract aggregate-likely mutant [20,21], we compared the double mutant (P24S/S31G) with WT and single one (P24S). To determine whether single and double mutants have different structural properties, we measured the intrinsic fluorescence of Trp of WT and single and double mutants. Since γD-crystallin is known to undergo conformational changes at acidic pH (pH 2.0) [37], we measured intrinsic fluorescence of Trp at pH 7.0 and 2.0. As shown in Figure 4a, no fluorescence difference between WT and two mutants was detected at pH 7.0, which is consistent with previous result [37]. However, intrinsic fluorescence of Trp of double mutant was significantly higher than those of WT and P24S single mutant at pH 2.0 (Figure 4b). The maximum emission wavelengths of intrinsic Trp fluorescence (λmax) was 338 nm for WT, while 4 nm red-shift for double mutant was observed. The intensity of intrinsic Trp fluorescence was also 1.3 times higher for the double mutant than WT. These results indicate that P24S/S31G double mutations of γD-crystallin alter conformational state and microenvironments more than P24S single mutation at the acidic pH. βB2-Crystallin is readily denatured at low concentrations of GdnHCl, while γD-crystallin is more resistant to denaturation by GdnHCl [16]. To investigate the stability differences between WT and the P24S and P24S/S31G mutants of γD-crystallin, we assessed intrinsic fluorescence changes by chemical denaturant GdnHCl (pH 7.4) in a dose dependent manner. As shown in Figure 4c, P24S/S31G mutant is more readily denatured at 2.0 M GdnHCl than WT and P24S mutant at 2.5 M.

To evaluate the long-term stability, the protein with a concentration of 5 mg/mL were incubated at 37 °C for 36 h and turbidity changes were measured with absorbance at 400 nm. In the same way with beta-crystallins. In contrast to the increase in turbidity due to aggregation of βB2-crystallin mutants, turbidity did not increase in γD-crystallin even after 36 h in both WT and mutant (Appendix A). Since gamma crystallin aggregates well in acidic conditions, the turbidity of WT and mutant was measured at 37 °C in a buffer condition of pH 2.0. The turbidity of mutant began to increase after 30 min and increased until 150 min to reach the maximum level, whereas turbidity of WT gradually increased as 90 min passed. This indicates that the mutant protein is more likely to aggregate in vitro. This result also observed that when mutants were overexpressed in mammalian cells, the mutants was small, but aggregation in the form of puncta was observed (Appendix A). These results indicate that S175G/H181Q mutation of βB2-crystallins reduced protein stability by affecting the protein folding and increased the tendency to be aggregated.

### 2.6. Amyloid Fibril Formation of γD-Crystallin Mutants Measured by ThT Fluorescence Spectroscopy

A previous study showed that the fibrillogenesis of human γD-crystallin could be induced by acidic pH [37]. Since P24S/S31G double mutant of γD-crystallin is unstable at acidic pH, we further investigated whether this double mutant more readily aggregates at acidic pH. We examined amyloidal structure of γD-crystallin mutants by employing ThioflavinT (ThT), a standard fluorescent dye that generates fluorescence by binding to amyloid structures. ThT is believed to bind to grooves formed on the amyloid fibrous surface by aligned side chain chains [38]. As shown in Figure 5a, at normal condition at 0 time point, all γD-crystallins are similar fluorescence intensities, however, during incubation at pH 2.0, increase in the rate of ThT fluorescence emission intensity is in the following order: P24S/S31G double mutant: P24S single mutant: and WT γD-crystallin. These results demonstrate that P24S/S31G double mutation promotes amyloid formation more readily than P24S mutant and WT γD-crystallin at acidic condition.

### 2.7. Effect of Incubation pH on the Tertiary Structure of γD-Crystallin Mutants Revealed by ANS Fluorescence Spectroscopy

To investigate the effects of double mutation of γD-crystallin on the tertiary structure at low pH, we examined the ANS spectra of WT and two mutants under acidic condition after various incubation times. Destabilization of the protein structure was assessed using ANS binding technique which measures the degree of exposure of hydrophobic regions. As shown in Figure 5b, ANS fluorescence intensities of all proteins increased during incubation time at pH 2.0. Increase in the rate of ANS fluorescence intensity is the largest with P24S/S31G double mutant, followed by P24S and WT. These results suggest that P24S/S31G double mutations induce more exposure of hydrophobic moieties to the surface than WT as well as P24S mutant, making the structure unstable at acidic condition.

### 2.8. Protein Modifications of WT and Mutant Recombinant γD-Crystallin Proteins

The structural changes of γD-crystallin in acidic condition cannot be examined with HDX-MS. Low pH is essential for pepsin digestion of protein for HDX-MS procedure. However, since γD-crystallin makes fibrils at low pH by aggregation [37], it is not possible to digest γD-crystallin aggregates at low pH with pepsin to examine the structural changes with HDX-MS. Therefore, we tried to determine the structural differences between WT and P24S/S31G mutant based on protein modifications, which vary depending on the structural states of the protein. We isolated WT and P24S/S31G mutant γD-crystallin recombinant proteins by SDS-PAGE and in gel-digestion with trypsin to obtain the peptides for analysis. We comprehensively analyzed the protein modifications of WT and P24S/S31G mutant γD-crystallin recombinant proteins employing peptide sequencing with nanoUPLC-ESI-q-TOF-MS/MS with SEMSA [39] which is a sensitive method for identifying low abundant modifications. Precursor ion chromatograms (XIC) of extracted precursor ions from both recombinant proteins are presented in Figure 6. All intensities of precursor ions were normalized by total ion chromatograms (TIC) of each sample run. Since protein modifications detected mainly in cataract lens are known as oxidation and deamidation [40,41], we assessed the protein modifications of WT and mutant γD-crystallins as shown in Table 2. Aging increases protein oxidation and cataract development is also promoted by oxidation of the lens [42,43]. Oxidations of Cys residues have been identified in cataract samples [42], and Met oxidation is known to be associated with cataractogenesis [44]. Of the six Cys residues of γD-crystallin, four Cys residues (Cys19, 42, 109 and 111) are predicted to be exposed to the surface of protein (Appendix A). Cys19 residue was not oxidized in WT, but was highly oxidized to sulfinic acid (16HYECSSDHPNLQPYLSR32, Δm = +32 Da) in P24S/S31G mutant (Figure 6a). Cys19 residue is present in the largest loop of the Greek key motif 1 where the mutation occurred. Therefore, the oxidation increase at Cys19 residue indicates the conformational change of Greek key motif 1 by mutation. Cys42 residue in double mutant was mainly not oxidized, but small faction of Cys42 was oxidized to thiosulfonic acid in double mutant (35SARVDSGCWMLYEQPNYSGLQYFLR59, Δm = +64 Da) (Figure 6b). Cys109 and Cys111 residues were detected in same tryptic peptide, and these are known to function as antioxidant by forming disulfide crosslinking in WT [45]. Cys109 residue was oxidized to dehydroalanine (Δm = −34 Da) and Cys111 was oxidized to sulfinic acid (Δm = +32 Da) in double mutant (Figure 6c). This indicates that Cys109-Cys111 disulfide formation does not work in the double mutant and lose their antioxidant function. In addition to Cys residue, Met residue is an oxidation sensitive amino acid. Met147 of γD-crystallin is known to be an important residue for protein solubility due to its proximity to the linker peptide and location at the interface between the *N*-terminal domain (NTD) and C-terminal domain (CTD) [46]. Met147 was also found to be more oxidized to mono- (143QYLLMPGDYR152, Δm = +16 Da), or di-oxidation (143QYLLMPGDYR152, Δm = +32 Da) in the double mutant, in particular the intensity of the double oxidation was found to be significantly increased compared to that of the unmodified peptide (Figure 6d–f). The results demonstrate that P24S/S31G double mutant is more readily oxidized than WT γD-crystallin. The results show that P24S/S31G mutant has different structure from WT γD-crystallin and is significantly aggregated in response to various stimuli causing cataract.

## 3. Discussion

In the present study, we identified 10 mutants of crystallin from proteomic analysis dataset of human cataract patients, and by comparing human lens samples of normal and cataract patients. Of these mutants, we focused on two cataract-specific double mutations, βΒ2-crystallin mutant, S175G/H181Q and γD-crystallin mutant, P24S/S31G. We have shown that double mutation of βΒ2-crystallin affects Greek key motif 4 structure in the C-terminus, which functions to maintain the stability and solubility of βΒ2-crystallin protein, and that double mutation of γD-crystallin induces structural changes and protein oxidation in Cys and Met residues that readily form amyloid fibril. These newly found mutations could serve as biomarkers of congenital cataract, as they appear to damage lens transparency by inducing the aggregation of β/γ-crystallin, decreasing its solubility and stability. These double mutations of crystallin occur adjacently in one tryptic peptide, which were in the critical loop of the Greek key motif. This report is the first study to investigate dynamic structural changes of crystallin mutants employing HDX-MS and to quantify protein oxidations of crystallin mutants combining with spectroscopic and biophysical analysis.

β/γ-Crystallins have four Greek key motifs in two domains with twisted order [4]. The Greek key motif is evolutionarily well conserved and crucial for the structure, function and stability of crystallin proteins. Mutations occurred in Greek key motifs are known as typical causes of cataract [3,27]. Nonsense mutant Q155X in βB2-crystallin deteriorates protein stability by unfolding the Greek key motif [15] and distorting only one Greek key motif of γD-crystalllin causes protein aggregation and the development of nuclear cataracts [27]. Our new double mutation sites are present in the largest loop connecting the distant β-sheets, which is an important factor in the stability of the Greek key motif [28]. In S175G/H181Q βB2-crystallin, the mutated amino acid is located in the largest loop between distant β-sheets in the Greek key motif 4. The sequence alignment indicated that S175 and H181 residues conserved in βB2-crystallin across species except bovine (Appendix A). This indicates that largest loop is important for preserving Greek key motifs. As shown in the Appendix A, Ser175 is in the helix and has the potential to form hydrogen bonds with the surrounding glycine in a largest loop. When it mutates to glycine, the helix structure can become unstable without forming hydrogen bonds. In addition, His181 is likely to form hydrogen bonds with surrounding Gln183. When it is substituted with glutamine, the hydrogen bond can be released and may not form the original loop shape. In this study, the results show that double mutations affect the stability, oligomerization, and folding of structures of βB2-crystallin. These changes were confirmed to analyze the dynamic structural changes employing HDX-MS. S175G/H181Q mutation in βB2-crystallin made packing core of Greek key motif 4 in C-terminus domain being loosened (Figure 3). Taken together, S175G/H181Q double mutant is unable to retain the Greek key motif 4, which impedes the achievement of βB2-crystallin folding, thereby causing cataracts. This is consistent with the results that misfolded βB2-crystallins form aggregates in both the *E. coli* and HeLa cells [35].

γD-Crystallin mutant P24S/S31G, unlike the double mutant of βB2-crystallin, is mutated in the Greek key 1 motif located in the *N*-terminus domain. The P24T and P24S mutants are known as cataract-causing mutants by increasing surface hydrophobicity that promote destabilization and aggregation [19,47]. New γD-crystallin mutant P24S/S31G we found is mutated not only in Pro24 but also in Ser31. Sequence alignment of γD-crystallins across species shows that Pro24 is about 60% conserved, while Ser31 is all conserved except bovine (Appendix A). Thus, it is possible that Ser31 is important for γD-crystallin structure. Indeed, P24S and P24T was identified in peripheral and nonnuclear cataract [27,47], while P24S/S31G mutation was found in nuclear cataract causing severe cataract in this study. P24S mutant can increase the surface hydrophobicity and reduce protein solubility [19]. Ser31 is presumed to form a hydrogen bond because the distance between nearby Arg15 and Arg32 is 4.44 and 6.60 Å, respectively in protein structure (Appendix A). Therefore, S31G mutant can destabilize the Greek key motif without hydrogen bonding. This is confirmed by comparing the structural properties of P24S/S31G double mutant with those of WT and P24S single mutant. P24S/S31G double mutant is more readily denatured than WT and P24S at acidic pH. However, the dynamic structural changes of γD-crystallin with HDX-MS could not be examined, because pepsin digestion in HDX-MS experiment should be performed at low pH. Instead, we tried to determine the structural difference between WT and P24S/S31G mutant based on the protein modifications. Protein oxidations at Cys and Met residues, which protect proteins in response to oxidative stress, were measured and compared between WT and P24S/S31G mutant. As shown in Figure 6, P24S/S31G mutant is more readily oxidized than WT γD-crystallin. Since β/γ-crystallin is vulnerable to enzyme-independent chemical modifications throughout lifespan [48,49], the results suggest that mutations make proteins being more vulnerable to oxidative stress because of structural changes.

In summary, this study identifies new two double mutations, S175G/H181Q in βB2- and P24S/S31G in γD-crystallin, are sufficient to promote aggregation with distorting the largest loops connecting the distant β sheets of Greek key motifs, possibly cause cataract. These new double mutants can be used for early diagnosis of cataract patients. Further studies are required to understand whether these mutants are involved in other crystallin-related diseases, i.e., γD-crystallin is known to be associated with prostate cancer [50].

## 4. Materials and Methods

### 4.1. MS/MS Dataset and Bioinformatics Analysis

Five normal human lenses (3-day and 2-, 18-, 35-, and 70-year old) and two nuclear cataract lenses (70-year old (Grade II, Pirie scale) and 93-year old (Grade III, Pirie scale)) were obtained from previous studies, where age-related changes in human crystallins were analyzed [24,25]. The detailed description of the data are provided in the previous studies [24]. In brief, the lenses were obtained from the Lions Eye Bank of Oregon with IRB approval, dissected within 12 h of death, graded for cataract severity, photographed, and frozen at −70 °C until use. Individual lenses were homogenized on ice in a 20 mM phosphate and 1.0 mM EDTA buffer (pH 7.0) using 1.0 mL of buffer per lens, and the homogenate was centrifuged at 20,000× *g* for 30 min to pellet the water-insoluble proteins. The water-insoluble pellet was re-suspended once in the same volume of homogenizing buffer, and again pelleted. The resulting pellet was re-suspended by brief sonication (5 s × 2) on ice, and the protein content of both soluble and insoluble fractions was determined in triplicate using a BCA assay (Pierce Biotechnology, Inc., Rockford, IL, USA). Portions (2.5 mg) of the samples were dried by vacuum centrifugation and stored at −70 °C until analysis.

Dried water-soluble or water-insoluble lens samples were dissolved in buffer containing 8.0 M deionized urea, 0.8 M Tris, 0.08 M methylamine, and 8.0 mM CaCl_2_ (pH 8.5). Cysteine residues were reduced and alkylated by successive treatment with DTT and iodoacetamide, and sequencing grade modified trypsin (Trypsin Gold from Promega, Madison, WI, USA) was added at a ratio of 1:25 protease/substrate, resulting in a dilution of the urea to a final 2.0 M concentration. Digestion occurred during an 18 h incubation at 37 °C with shaking. Following digestion, formic acid was added to a final concentration of 5%, and peptides were solid-phase-extracted using Sep-Pak Light cartridges (Millipore, Billerica, MA, USA). The peptides were separated by SCX chromatography and were analyzed on a Thermo LCQ Classic instrument (Thermo Electron, San Jose, CA, USA). Their MS/MS data sets were downloaded from public repository (MassIVE, MSV000078532) and the numbers of MS/MS spectra for all lens datasets are shown in Appendix A. The amount of water-insoluble material from the 3-day old lens was negligible and not analyzed.

The MS/MS spectra were searched in an unrestrictive way using MODplus [51] because the human lens tissue is often substantially modified post-translationally as it ages. The MODplus search allowed 946 chemical modifications and 360 amino acid substitutions (modifications in −150 and +350 Da range listed in Unimod, July 2018) as variable modifications during search. The other parameters were set as follows: precursor mass tolerance ±2.5 Da, fragment mass tolerance ±0.5 Da, trypsin as enzyme, the number of enzymatic termini 1/2, any number of missed cleavages, any number of modifications/peptide, Carbamidomethyl (Cys) as a fixed modification, SwissProt human database (August 2019 release, 42,605 entries including common contaminant proteins). The protein database was appended with pseudo-shuffled decoy sequences for FDR (False Discovery Rate) control using target-decoy [52].

PSMs (peptide spectrum matches) with a minimum peptide length of 8 were identified at FDR 1% and the numbers of identified PSMs are shown in Appendix A. The fractions of crystallin proteins in each lens sample identified by MODplus are also shown in Appendix A. To conservatively call amino acid mutations from identified PSMs, they were accepted only if there were no possible chemical modifications matching the delta masses, the absolute size of delta masses were larger than 3 Da, and MODplus quality scores of their PSMs were higher than 0.1. As a result, total 112 mutations were identified from all lens samples (Appendix A). 85 mutations were observed in a single sample, 13 in both normal and cataract samples, 2 in some normal samples but never in cataract samples, and 12 in all cataract samples but never in other normal samples. The numbers of mutations observed for each crystallin and sample are shown in Appendix A, respectively. More mutations were observed in cataract samples and Table 1 shows mutated peptides containing 12 cataract-specific mutations (i.e., observed in both 70- and 93-year cataract lens samples but never in other normal lenses). Such occurrence can hardly happen by chance. For the identified 112 mutations across seven samples, the probability of observing at least 12 mutations only in those two samples by chance is 9.264 × 10^−10^.

### 4.2. Purification of Recombinant Proteins

For protein purification, the constructs encoding pGEX-4t-1 vector and a CRBB2, CRYGD cDNA clone were obtained from human cDNA library as a template. All plasmid constructs were confirmed by DNA sequencing. Mutant CRBB2 S175G/H181Q, and CRYGD P24S/S31G were generated using the Quick-Change site-directed mutagenesis kit (Agilent Technologies, Santa Clara, CA, USA) according to the manufacturer’s protocol using pGEX-4t-1 CRBB2 and CRYGD as a template. The primers used for mutagenesis are: CRBB2 S175G_sense, 5′-ACTACAAGGACAGCGGCGACTTTGGGGCC-3′; CRBB2 S175G_antisense, 5′-GGCCCCAAAGTCGCCGCTGTCCTTGTAGT-3′; CRBB2 H181Q_sense, 5′-CTT TGG GGC CCC TCA GCC CCA GGT-3′; CRBB2 H181Q_antisense, 5′-ACC TGG GGC TGA GGG GCC CCA AAG-3′; CRYGD P24S_sense, 5′-CAG CAG CGA CCA CAG CAA CCT GCA GCC C-3′; CRYGD P24S_antisense, 5′-GGG CTG CAG GTT GCT GTG GTC GCT GCT G-3′; CRYGD S31G_sense, 5′-TGC AGC CCT ACT TGG GCC GCT GCA ACT C-3′; CRYGD S31G_antisense, 5′-GAG TTG CAG CGG CCC AAG TAG GGC TGC A-3′. Double mutants (CRBB2 S175G/H181Q, CRYGD P24S/S31G) were sequentially generated using single point-mutated DNA as a template. All plasmid constructs were confirmed by DNA sequencing. A pGEX-4t-1 plasmid carrying the human CRBB2, CRYGD gene was expressed in BL21 (DE3) *E. coli* cells. The bacteria were cultured in LB medium at 37 °C, and recombinant fusion protein production was induced with 0.25 mM were isopropyl-β-d-thiogalactopyranoside (IPTG). After 4 h of additional incubation at 37 °C, the cells were harvested and lysed by vortexing with lysozyme containing lysis buffer (lysozyme, protease inhibitor, triton X-100 in PBS (140 mM NaCl, 2.7 mM KCl, 10 mM Na_2_HPO_4_, pH 7.4)) on ice and sonicated. The soluble protein fraction was recovered by centrifugation at 13,000× *g* for 30 min at 4 °C. The GST-fused recombinant proteins in the supernatant were purified by chromatography on a glutathione-agarose column followed by washing, and equilibration by elution PBS buffer. To cleave crystallin from the GST-crystallin, the beads were incubated overnight at room temperature with thrombin in crystallin elution buffer. After 16 h, the purified crystallin was eluted and protein concentration was quantified with the BCA protein assay.

### 4.3. Measurement of Intrinsic Fluorescence of Tryptophan and Tyrosine

All spectroscopic experiments were carried out at 25 °C using protein samples prepared in PBS buffer (140 mM NaCl, 2.7 mM KCl, 10 mM Na_2_HPO_4_, pH 7.4). The fluorescence spectra were recorded on SpectraMax i3x spectrophotometer (Molecular Devices, San Jose, CA, USA) with a slit width of 9 nm for excitation and 14 nm for emission. The protein concentration was 0.2 mg/mL for the fluorescence measurements. The excitation wavelength of the intrinsic Trp and Tyr fluorescence was 280 and 295 nm respectively. The ANS (8-anilino-1-naphthalenesulfonic acid) fluorescence was determined at excitation wavelength of 380 nm. For ANS fluorescence measurement at 470 nm, samples were prepared by mixing the protein with ANS stock solution to final molar ratio of 75:1 (ANS:protein) and equilibrating in the dark for 30 min at 25 °C. Each experiment was triplicated with independent samples.

### 4.4. Protein Denaturation Induced by GdnHCl

Protein unfolding induced by guanidinium hydrochloride (GdnHCl) was performed by incubating the WT and mutated crystallins in phosphate buffer containing various concentrations of GdnHCl (pH 7.4) ranging from 0 to 4.0 M 16 h at 25 °C. The final protein concentration was 0.2 mg/mL. The unfolded samples were used for intrinsic Trp and ANS fluorescence measurements to monitor the structural changes and appearance of aggregates during unfolding.

### 4.5. Measurement of Protein Oligomer States with Size Exclusion Chromatography

Protein (450 µg/100 µL) was loaded on a SuperdexTM 200 HR 10/30 column (GE Healthcare, Chicago, IL, USA) equilibrated with PBS. Samples were eluted at a flow rate of 0.5 mL/min and detected by UV absorbance at 280 nm. Proteins used as molecular weight markers were albumin (66.4 kDa) and α-lactalbumin (14.2 kDa).

### 4.6. Protein Turbidity Experiments

All spectroscopic experiments were carried out at 25 °C using protein samples prepared in PBS buffer. The turbidity of the samples was monitored by the absorbance at 400 nm (A_400_) using an Agilent 8453 UV-Visible spectrophotometer (Agilent, Santa Clara, CA, USA) with quartz glass high performance cuvette (Hellma Analytics, Munich, Germany). The protein concentrations was 5 mg/mL and 2.5 mg/mL for the experiment at pH 7.0 and pH 2.0, respectively.

### 4.7. Hydrogen/Deuterium Exchange Mass Spectrometry (HDX-MS)

HDX-MS experiment was performed as reported [23]. βB2-crystallin (1 µL, 1 mg/mL) was incubated with 9-fold D_2_O at 25 °C for the following periods of time: 10, 60, 300, 1800, or 10,800 s. The deuterium labeling reaction was performed by quenching with 2.5 mM Tris (2-carboxyethyl) phosphine (TCEP), formic acid, pH 2.3. For protein digestion, 1 µg of porcine pepsin was added to each quenched protein sample and incubated on ice for 3 min before injection. Peptic peptides were desalted on C18 trap column cartridge (Waters, Milford, MA, USA) and gradient eluted form 8% CH_3_CN to 40% CH_3_CN, 0.1% formic acid on 100 µm i.d. × 100 mm analytical column, 1.7 µm particle size, BEH130 C18, (Waters, Milford, MA, USA) for 7 min. The trap, analytical column and all tubing were immersed in an ice bath to minimize deuterium back-exchange. Gradient chromatography was performed at a flow rate 0.5 µL/min and was sprayed on line to a nanoAcquity^TM^/ESI/MS (SYNAPT^TM^HDMS^TM^) (Waters, Milford, MA, USA). The extent of deuterium incorporation was monitored by the increase in mass of the isotope distribution for each identified peptide, and calculated using Microsoft Excel. The theoretical maximum deuterium incorporation value was calculated for each peptide based on the number of exchangeable amides. Each experiment was triplicated with independent samples.

### 4.8. Thioflavin T (ThT) Fluorescence Measurement

The stock solution of ThT at a concentration of 20 mM was prepared in ethanol. PBS buffer was used to dissolve ThT to a final concentration of 20 µM. 40 µL (1 mg/mL) of protein samples taken at different times were added to 960 µL of ThT solution (20 µM) and briefly mixed. The ThT fluorescence emission intensity at 485 nm was monitored using the excitation wavelength of 440 nm. Each experiment was triplicated with three independent samples.

### 4.9. PTM Analysis of γD-Crystallin Using NanoUPLC-ESI-q-TOF Tandem MS

Recombinant WT and mutant γD-crystallin protein (1 µg) were separated on SDS-PAGE and stained with Coommassie blue. In these experiments, to prevent oxidation during sample preparation, the alkylating agent *N*-ethylmaleimide (NEM) was applied to all experiments and the experiment was conducted two times. The gel bands of differentially expressed proteins were destained and digested with trypsin, and the resulting peptides were extracted as previously described [23]. The peptides were separated using trap column cartridge, injected into a C18 reversed-phase analytical column with an integrated electrospray ionization PicoTip^TM^ using nanoAcquity^TM^ UPLC/ESI/q-TOF MS/MS (SYNAPT^TM^ G2Si; Waters Co.). Peptide solutions were injected into a column and eluted by a linear gradient of 5–40% buffer B (ACN/formic acid; 100:0.1, *v*/*v*) with buffer A (Water/formic acid; 100:0.1, *v*/*v*) over 60 min. The mass spectrometer was programmed to record scan cycles composed of one MS scan followed by MSMS scans of the 10 most abundant ions in each MS scan. Tandem MS (MS/MS) spectra were matched against amino acid sequences in the SwissProt human database using Modplus [51].

### 4.10. Confocal Microscopy

Confocal experiment was performed as reported [53]. Hela cells were plated on the glass coverslip 24 h before transfection. Cells were transfected with 1 µg of Flag, Flag CRBB2/CRYGD WT or mutant. After 24 h, cells were fixed with 4% paraformaldehyde. Cell were permeabilized with 0.1% Triton X-100, blocked with blocking solution (3% bovine serum albumin; 0.2% Tween 20 and 0.2% gelatin) and incubated with anti-Flag (20 µg/mL) primary antibodies for 2 h at 37 °C. Alexa Fluor 488 conjugated goat anti-mouse (1:100) secondary antibodies (Invitrogen, Carlsbad, CA, USA) were used to visualize Flag-crystallin. Coverslips were mounted using antifading solution containing DAPI (Molecular Probes) and images were obtained using LSM 880 Airy scan confocal microscope (Zeiss, Oberkochen, Germany).

## Figures and Tables

**Figure 1 ijms-21-06504-f001:**
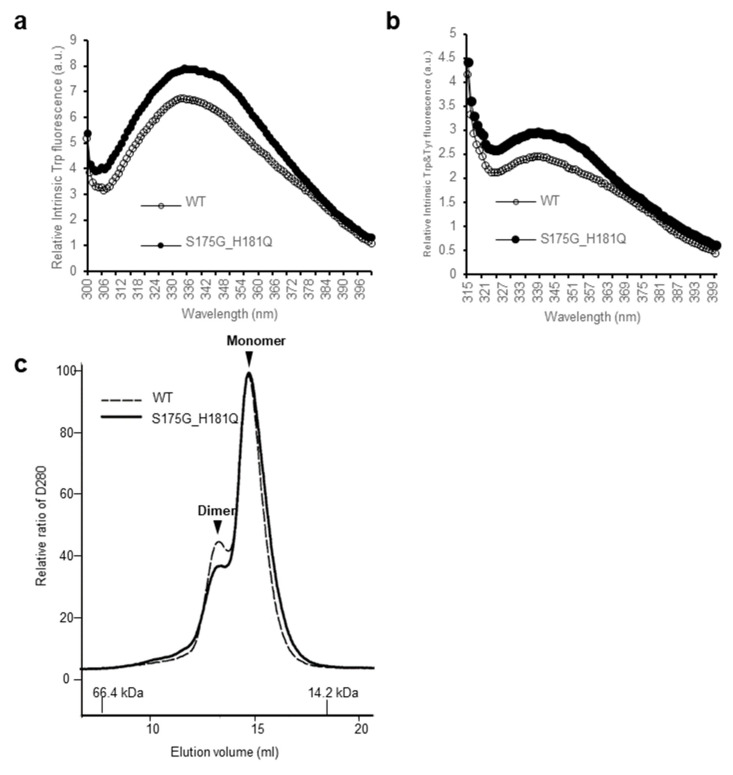
Effect of the double mutations (S175G/H181Q) on the secondary, tertiary structures and oligomerization of βB2-crystallin. (**a**) Intrinsic Trp fluorescence excited at 295 nm. (**b**) Intrinsic Trp and Tyr fluorescence excited at 280 nm. The proteins were prepared in PBS buffer with concentration of 0.5 mg/mL. The fluorescence intensity was recorded for every 1 nm for WT (opened circles) and the S175G/H181Q mutant (closed circles). (**c**) Size exclusion chromatograms of WT and S175G/H181Q mutant βB2-crystallin using Superdex^TM^ 200 HR 10/30 column. 450 µg of WT protein (black dotted line) and the S175G/H181Q mutant (black line) were loaded on column equilibrated with PBS buffer. Albumin (66.4 kDa) and α-lactalbumin (14.2 kDa) were used as molecular weight markers.

**Figure 2 ijms-21-06504-f002:**
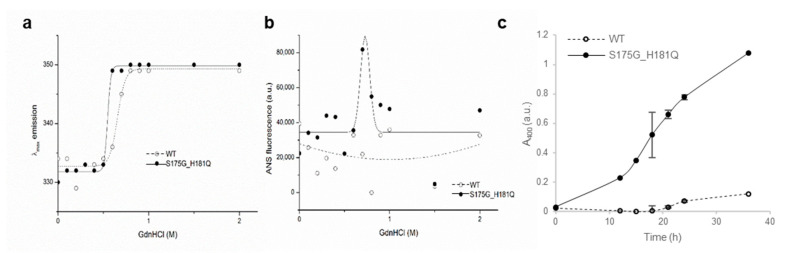
Denaturation of βB2-crystallin protein with various concentrations of GdnHCl. (**a**) Transition curves of protein equilibrium unfolding monitored by Emax of Trp intrinsic fluorescence (Ex: 280 nm). Data for each protein samples were fitted with a Boltzmann curve using Origin 8.5 (OriginLab Corporation, Northampton, MA, USA). The Emax did not change at GdnHCl concentrations above 1 M. (**b**) Hydrophobic exposure during GdnHCl-induced unfolding was monitored by ANS fluoresescence intensity at 470 nm. Data for each proteins were fitted with a GaussAmp curve using Origin 8.5. (**c**) Long-term stability revealed by incubating 5.0 mg/mL proteins at 37 °C continuously, and turbidity changes at 400 nm were measured.

**Figure 3 ijms-21-06504-f003:**
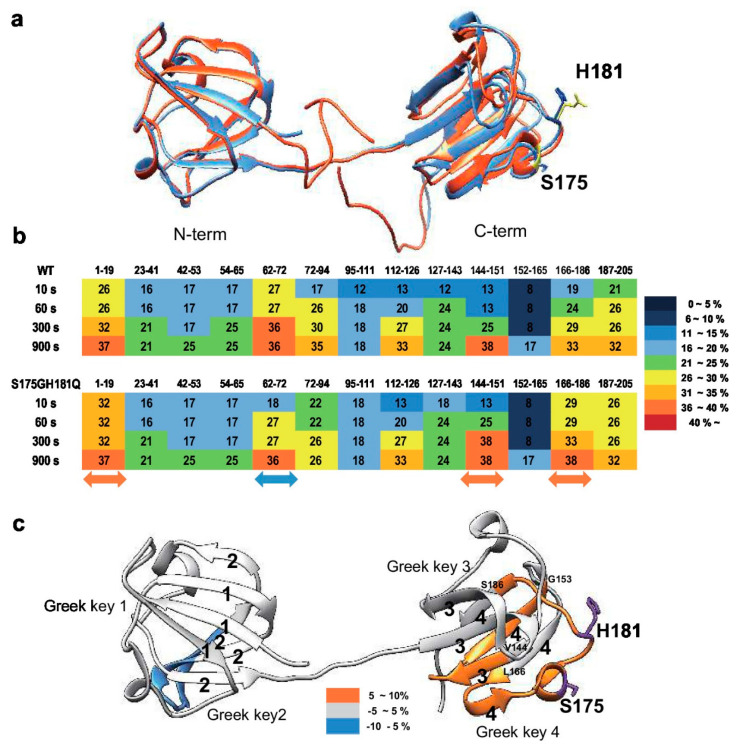
Structural analysis between wild-type and S175G/H181Q mutant of βB2-crystallin. (**a**) The predicted tertiary structure of the S175G/H181Q mutant. Phyre2 structure predictions for the mutant is shown in red ribbon. WT is shown in blue ribbon (PDB ID code: 1YTQ). (**b**,**c**) Structural changes in WT and the mutant of βB2-crystallin analyzed by hydrogen/deuterium exchange mass spectrometry. (**b**) Deuterium exchange rate (%) of WT and S175G/H181Q mutant during incubation with D_2_O. Significantly changed representative peptides were marked by colored arrows. The corresponding deuterium exchange levels for each peptide in percent are given on the right. (**c**) Changes in hydrogen/deuterium exchange-mass spectrometry data of the mutant of βB2-crystallin compared to the WT onto the structures of human βB2-crystallin structure (PDB ID: 1YTQ). Percentage difference of deuterium incorporation in HDX-MS between WT and S175G/H181Q mutant is colored according to the key.

**Figure 4 ijms-21-06504-f004:**
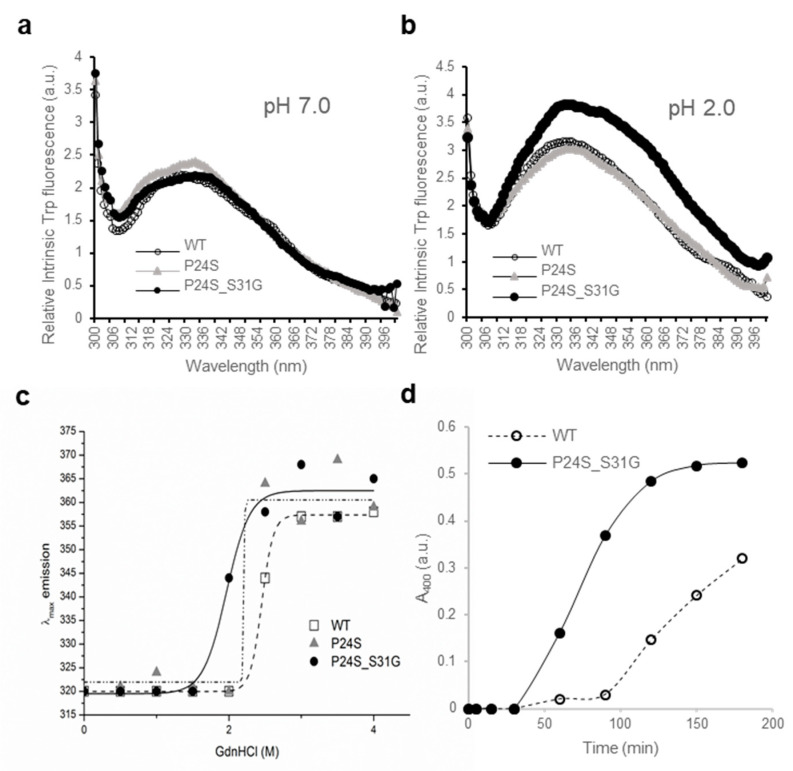
Effect of the double mutations (P24S/S31G) on the secondary and tertiary structures of γD-crystallin by spectroscopic methods. Intrinsic Trp fluorescence excited at 295 nm (**a**) at pH 7.0, and (**b**) at pH 2.0. The proteins were prepared in PBS buffer with concentration of 0.5 mg/mL. The fluorescence intensity was recorded for every 1 nm for WT (opened circles), the P24S mutant (gray triangles) and the P24S/S31G mutant (closed circles). (**c**) Effect of P24S/S31G double mutations on the folding and stability of γD-crystallin during denaturation induced by GdnHCl. WT and mutants were incubated for 16 h with various concentrations of GdnHCl, and its intrinsic Trp fluorescence intensities were monitored from 310 nm to 400 nm. Transition curves of protein equilibrium unfolding monitored by Emax of Trp fluorescence. Data for each protein samples were fitted with a Boltzmann curve using Origin 8.5. The final protein concentration was 0.2 mg/mL. (**d**) Long-term stability revealed by incubating 2.5 mg/mL proteins at pH 2.0 and 37 °C continuously, and turbidity data were measured at given intervals.

**Figure 5 ijms-21-06504-f005:**
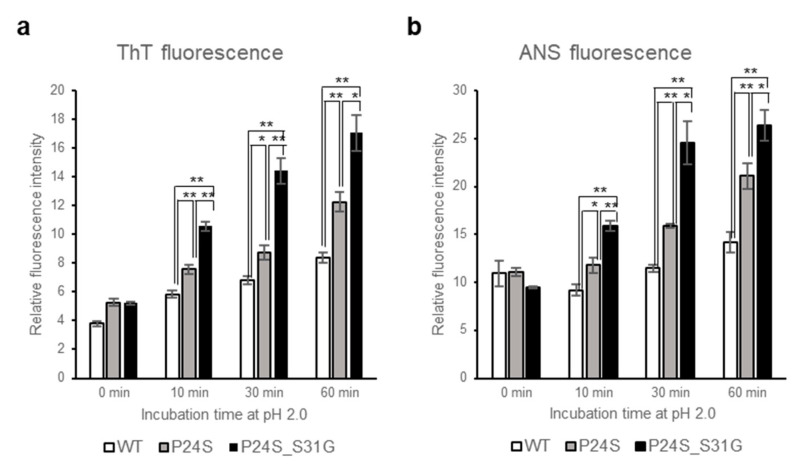
Effect of P24S/S31G double mutations on the fibril formation and tertiary structure of γD-crystallin in acidic pH. (**a**) ThioflavinT fluorescence emission intensity measurement (λ_Ex_ = 440 nm, λ_Em_ = 485 nm) of WT, P24S, and P24S/S31G mutant γD-crystallin as a function of incubation time. Samples were incubated at pH 2.0 for indicated time. (**b**) ANS fluorescence intensity measurement of WT, P24S and P24S/S31G mutant γD-crystallin. ANS fluorescence emission intensity of γD-crystallin sample as a function of incubation time are represented with WT (white bar), the P24S mutant (gray bar) and the P24S/S31G mutant (black bar). The final concentration was 0.04 mg/mL. Data points are presented as the means ± standard deviations (S.D.) of at least 3 independent measurements in each Figure. * *p* < 0.05; ** *p* < 0.01 versus control.

**Figure 6 ijms-21-06504-f006:**
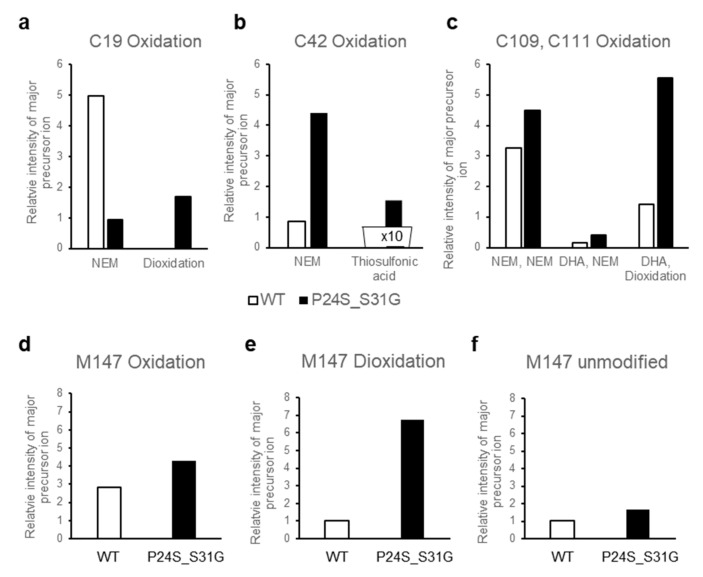
The relative intensity of precursor ion peptide containing modifications in WT and P24S/S31G mutant γD-crystallin (**a**–**f**). The relative intensity of precursor ion unmodified and oxidized peptide including (**a**) Cys19, (**b**) Cys42, (**c**) Cys109 and 111, (**d**–**f**) Met147 with WT (white bar) and P24S/S31G mutant (black bar). NEM means NEM (*N*-ethylmaleimide) adducts of Cys residue that is not oxidized. Total ion chromatogram (TIC) intensity of sample as control normalize each intensity of modifications. Each experiment was repeated twice.

**Table 1 ijms-21-06504-t001:** The list of novel mutations from nuclear cataract human lens samples.

Protein	From	To	Peptide
Alpha-crystallin B chain	57	69	APSW(F→I/L)DTGLSEMR
Alpha-crystallin B chain	124	149	IPADVDPL(T→A)ITSSLSSDGVLTVNGPR
Beta-crystallin A4	178	192	EWGSHA(P→Q)TFQVQSIR
Beta-crystallin B1	93	110	SIIVS(A→T)GPWVAFEQSNFR
Beta-crystallin B1	215	230	HWNEWGAFQPQMQ(S→G)LR
Beta-crystallin B2	122	140	ME(I→V)IDDDVPSFHAHGYQEK
Beta-crystallin B2	169	188	GDYKDS(S→G)DFGAP(H→Q)PQVQSVR
Beta-crystallin B3	56	71	VGSIQVESGPWLAFE(S→R)
Beta-crystallin B3	116	127	LHLFENPAF(S→G)GR
Gamma-crystallin D	16	32	HYECSSDH(P→S)NLQPYL(S→G)R

**Table 2 ijms-21-06504-t002:** Post-translational modifications detected in recombinant protein WT and double mutant (P24S/S31G) of γD-crystallin. Peptides with the highest ion scores for each spot were tabulated. Modified residues are highlighted in bold. MS/MS spectra of modified peptides are presented in Appendix A.

Residue	Modification	Start–End	Mass (*m*/*z*) Experimental	Mass Theoretical	Delta Mass (Da)	Peptide Sequence
C19	Nethylmaleimide	16–32	724.3238(3+)	2169.9496	−0.0042	HYECSSDHPNLQPYLSR
	Dioxidation	16–32	679.9601(3+)	2036.8585	−0.0062	HYECSSDHSNLQPYLGR
C42	Nethylmaleimide	38–59	932.0887(3+)	2793.2443	−0.0124	VDSGCWMLYEQPNYSGLQYFLR
	Trioxidation	35–59	758.5914(4+)	3030.3365	−0.0275	SARVDSGCWMLYEQPNYSGLQYFLR
	Thiosulfonic acid	35–59	762.5911(4+)	3046.3353	−0.044	SARVDSGCWMLYEQPNYSGLQYFLR
C109/ C111	Nethylmaleimide/Nethylmaleimide	100–115	1062.9352(2+)	2123.8558	−0.004	GQMIEFTEDCSCLQDR
	Cys->Dha/ Nethylmaleimide	100–115	983.4172(2+)	1964.8198	−0.0046	GQMIEFTEDCSCLQDR
	Cys->Dha/ Dioxidation	100–115	936.8794(2+)	1871.7442	−0.0046	GQMIEFTEDCSCLQDR
M147	Oxidation	143–152	636.307(2+)	1270.5994	−0.0022	QYLLMPGDYR
	Dioxidation	143–152	644.3046(2+)	1286.5946	−0.0019	QYLLMPGDYR

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
