# Peer review of "Cataract-Associated New Mutants S175G/H181Q of βΒ2-Crystallin and P24S/S31G of γD-Crystallin Are Involved in Protein Aggregation by Structural Changes"

_ijms, 2020, doi:10.3390/ijms21186504_

Round 1
Reviewer 1 Report
In the study, Song et al screened mutations in crystallins using proteomic method, and they identified 10 new cataract-specific mutations. Among them, the authors focused two double mutants, S175G/H181Q of beta-B2-crystallin and P24S/S31G of gamma-D-crystallin, for its mutation sites are located in crucial loops of Greek key motif. These mutants were further investigated and found that their structures are less stable compared to wild-type proteins, and the authors concluded that the decreased stability induce the aggregation of crystallins and results in cataracts. This reviewer think this research will contributes firmly to understanding cataracts and investigating their mechanisms.
However, this reviewer think this manuscript has a major concern and two minor concerns as follows. These points seem to impair the accuracy of the paper, and some revision should be done.
The major concern: lack of the evidences of induced aggregation by mutations.
The authors measured the decreased structural stability of the mutants by several types of experiments, but did not show the direct evidence of the aggregation caused by mutations. In general, protein aggregation can be evaluated using Dynamic Light Scattering, Laser Diffraction, Size exclusion chromatography (SEC), etc. In this paper, the authors used SEC only for S175G/H181Q mutant analysis, but the data shows increased monomerization by mutation rather than aggregation. And also, they did not apply any assessment of aggregation to the other mutant (P24S/S31G). Thus, it is impossible to say the mutations they found cause protein aggregation. Without direct observation, I think it can only be concluded that these mutants cause structural changes that “may” lead to aggregation.
The minor concerns:
1: The reason to avoid HDX-MS on P24S/S31G mutant in line 252-254.
Among the two mutants studied, they did not perform HDX-MS on P24S/S31G mutant, but the reason seems to be not clear. The authors described the reason in line 252-254. However, this reviewer can not understand why the instability of this mutant at low pH interferes with the pepsin digestion at low pH. Please explain more specifically, and
this should help more audience understand the unavoidable circumstances where two mutants could not be analyzed in the same way.
2: Fig4c.
The measurement of P24S mutant seems not to be performed at the [GdnHCl] from 1 to 2.5 M, and this makes difficult to assume precisely the threshold of [GdnHCl] that cause the denaturation. However, the authors concluded that to be 2 M, the threshold concentration of wild-type gamma-D-crystallin. It seems to be necessary to add data points or to change the interpretation of the data.
*an additive concern: please explain the abbreviations NTD and CTD in line 281.
Author Response
Reviewer #1 (Remarks to the Author):
In the study, Song et al screened mutations in crystallins using proteomic method, and they identified 10 new cataract-specific mutations. Among them, the authors focused two double mutants, S175G/H181Q of beta-B2-crystallin and P24S/S31G of gamma-D-crystallin, for its mutation sites are located in crucial loops of Greek key motif. These mutants were further investigated and found that their structures are less stable compared to wild-type proteins, and the authors concluded that the decreased stability induce the aggregation of crystallins and results in cataracts. This reviewer think this research will contributes firmly to understanding cataracts and investigating their mechanisms.
However, this reviewer think this manuscript has a major concern and two minor concerns as follows. These points seem to impair the accuracy of the paper, and some revision should be done.
The major concern: lack of the evidences of induced aggregation by mutations.
The authors measured the decreased structural stability of the mutants by several types of experiments, but did not show the direct evidence of the aggregation caused by mutations. In general, protein aggregation can be evaluated using Dynamic Light Scattering, Laser Diffraction, Size exclusion chromatography (SEC), etc. In this paper, the authors used SEC only for S175G/H181Q mutant analysis, but the data shows increased monomerization by mutation rather than aggregation. And also, they did not apply any assessment of aggregation to the other mutant (P24S/S31G). Thus, it is impossible to say the mutations they found cause protein aggregation. Without direct observation, I think it can only be concluded that these mutants cause structural changes that “may” lead to aggregation.
A: βB2-crystallin forms oligomer when it has a proper tertiary structure. Previous study demonstrates that the R188H mutant, which is known to be aggregated by structural changes, does not form an oligomeric equilibrium (ref. 35). In this study, the S175G/H181Q mutant also showed the same trend, confirming that it did not form oligomer well. We corrected the method title to "Measurement of protein oligomer states with size exclusion chromatography”
As suggested, we performed two additional experiments to examine the aggregation of two mutants. We found that the mutants are more readily aggregated than the WT in vitro and in vivo. We added the Figure 2c and 4d, showing the turbidity increases of mutant proteins, and Fig. S5, observing the protein aggregation increase in the cells transfected with mutants.
The minor concerns:
1: The reason to avoid HDX-MS on P24S/S31G mutant in line 252-254.
Among the two mutants studied, they did not perform HDX-MS on P24S/S31G mutant, but the reason seems to be not clear. The authors described the reason in line 252-254. However, this reviewer can not understand why the instability of this mutant at low pH interferes with the pepsin digestion at low pH. Please explain more specifically, and this should help more audience understand the unavoidable circumstances where two mutants could not be analyzed in the same way.
A: We explained more clearly to understand why we cannot use HDX-MS for structural analysis of P24/S31G mutant. Low pH is essential for pepsin digestion of protein for HDX-MS procedure. However, since γD-crystallin makes fibril at low pH by aggregation (ref 37), it is not possible to digest γD-crystallin aggregates at low pH with pepsin to examine the structural changes with HDX-MS.
2: Fig4c.
The measurement of P24S mutant seems not to be performed at the [GdnHCl] from 1 to 2.5 M, and this makes difficult to assume precisely the threshold of [GdnHCl] that cause the denaturation. However, the authors concluded that to be 2 M, the threshold concentration of wild-type gamma-D-crystallin. It seems to be necessary to add data points or to change the interpretation of the data.
A: We corrected Fig 4c to add the line of the P24S mutant. We clearly show that the P24S mutant is stable up to 2 M [GdnHCl] and more denatured at 2.5 M than the WT. This also indicates that the double mutant is more structurally unstable than the P24S mutant.
*an additive concern: please explain the abbreviations NTD and CTD in line 281.
A: As suggested, we explained the abbreviations NTD and CTD.

Reviewer 2 Report
The authors have analyzed proteomics dataset from two cataract patients and they are reporting two new nuclear cataract-associated mutants in βB2- and γD-crystallins. The mutations are S175G/H181Q on βB2-crystallin and P24S/S31G on γD-58 crystallin. Following expression and purification of double mutants the investigators examined the structural changes in the proteins using spectroscopic and mass spectrometric analysis. The results showed appreciable structural changes in βB2-crystallin S175G/H181Q 61 mutant. They also found structural changes in γD-crystallin P24S/S31G 63 mutant in acidic condition. Furthermore, the gamma mutant was more susceptible to oxidation. Based on these observations the authors are of the opinion that the mutations might be promoting cataractogenesis, and they could serve as biomarkers of cataract development.
The data presented partially supports the authors view. There are several minor and few major concerns regarding the study, presentation of the data and interpretation. They are summarized below.
The data sets from a 2006/2007 study were analyzed by the authors. The data sets from 7 samples showed unusually large number of mutations, 112 in total. The authors should clearly show the publicly available data set access site in the manuscript. A major surprise is that the two cataract samples (70 and 93 yr) showed 37 mutations each! This many number of mutations are significantly higher than one can expect in a normal or diseased lens sample. The authors also state in line 407 (page 12) that they found 85 mutations in single sample! This is likely an error. Twelve cataract-specific mutations were found in 2 cataract lenses. Thus, the number of mutations the authors report/lens are much higher than those observed in all previous studies. Additionally, which of the mutations are silent and which are drive the lens into cataract phenotype is unclear. Since those cataract lenses were extracted from patients at 70 and 93 yr it is very surprising that the lenses remained transparent for several decades, in spite of having 12 cataract specific mutations in each. The results of in vitro studies included in the manuscript demonstrate that the two double mutations destabilize beta B2- and gamma D-crystallins and induce protein oxidation. Therefore, it is difficult to comprehend that the lenses obtained from 70 and 93 yr patients remained transparent several decades and have to be extracted late in the life! From the data included in the manuscript it does not appear that the authors were able to look at the MS/MS data to unequivocally prove that normal and cataract lenses had so many mutations, summarized in Table S3 and S4. Additionally, it is very surprising that the investigators who originally analyzed those lens proteins and prepared data sets did not recognize such large number of mutations.
Results:
Lines 114 and 233: satatements regarding the secondary/tertiary structural changes. The authors should perform CD spectroscopy of purified WT and double mutants and analyze α-helical, β-sheet and random structure content and include the results in the Suppl section.
Fig 2 a and 4 C: It will be good to estimate the ∆G of the protein unfolding and include in the results.
Fig 5 b: It is not clear whether the unfolding is complete at 60 mins. Data from an additional time point should be included to confirm this.
Oxidation of gamma D-mutants: From the data shown in Fig 6 and Table 2 it is clear that the double mutant is more susceptible to oxidation. It is also known that gamma crystallins are more susceptible to oxidation and oxidation can result in protein conformation changes, including the exposure of hydrophobic regions. Data presented in Fig 4 shows that the double mutant displays higher degree of conformation change and decreased stability. On examining the methods section it appears that the double mutant (P24S/S31G) was not kept in buffers containing reducing agent to prevent oxidation of Cys in gamma protein. Therefore, it is possible that the proteins that showed increased ThT binding and ANS binding were oxidized and the conformation change observed was due to protein oxidation rather than mutation by itself. This should be clarified.
Solubility of mutants: The authors have not performed solubility tests for the proteins or not shown any data that demonstrates aggregation and precipitation of mutants. It will be good to provide this information in the manuscript. If the proteins have decreased solubility, how they were purified/recovered.
Methods:
Lines 370 to 381- The write up is similar to the method section in ref 24. This section should be presented between quotes with proper acknowledgement. Otherwise, the authors need to re-write this part of the manuscript.
Suppl section:
Figure S4: Realign “ – “ and “+” H2O2” above the gel. What are those bands above dimers in samples without ME ? The so called monomers in lanes showing samples without ME are seen as double bands. Why ?
Figure S6 legend- correct “spetra” as “spectra”
Figure S8: What is the source of the data ? If this is a replot, based on literature data appropriate reference and permission to reproduce has to be stated.
Table S4, a and b: The authors claim that they have identified 37 mutations in 70 and 93 yr old cataract lenses. It is very surprising that in spite of so many mutations those lenses remained transparent (?) for 7 to 9 decades of life since the authors also show that the double mutants of beta B2 and gamma D are unstable, aggregate and likely cataractogenic.
Author Response
Reviewer #2 (Remarks to the Author):
The authors have analyzed proteomics dataset from two cataract patients and they are reporting two new nuclear cataract-associated mutants in βB2- and γD-crystallins. The mutations are S175G/H181Q on βB2-crystallin and P24S/S31G on γD-58 crystallin. Following expression and purification of double mutants the investigators examined the structural changes in the proteins using spectroscopic and mass spectrometric analysis. The results showed appreciable structural changes in βB2-crystallin S175G/H181Q 61 mutant. They also found structural changes in γD-crystallin P24S/S31G 63 mutant in acidic condition. Furthermore, the gamma mutant was more susceptible to oxidation. Based on these observations the authors are of the opinion that the mutations might be promoting cataractogenesis, and they could serve as biomarkers of cataract development.
The data presented partially supports the authors view. There are several minor and few major concerns regarding the study, presentation of the data and interpretation. They are summarized below.
The data sets from a 2006/2007 study were analyzed by the authors. The data sets from 7 samples showed unusually large number of mutations, 112 in total. The authors should clearly show the publicly available data set access site in the manuscript.
A: We corrected to describe that the data set can be downloaded from MassIVE (https://massive.ucsd.edu) with the accession MSV000078532 in line 388 in page 12.
A major surprise is that the two cataract samples (70 and 93 yr) showed 37 mutations each! This many number of mutations are significantly higher than one can expect in a normal or diseased lens sample. The authors also state in line 407 (page 12) that they found 85 mutations in single sample! This is likely an error. Twelve cataract-specific mutations were found in 2 cataract lenses. Thus, the number of mutations the authors report/lens are much higher than those observed in all previous studies. Additionally, which of the mutations are silent and which are drive the lens into cataract phenotype is unclear. Since those cataract lenses were extracted from patients at 70 and 93 yr it is very surprising that the lenses remained transparent for several decades, in spite of having 12 cataract specific mutations in each. The results of in vitro studies included in the manuscript demonstrate that the two double mutations destabilize beta B2- and gamma D-crystallins and induce protein oxidation. Therefore, it is difficult to comprehend that the lenses obtained from 70 and 93 yr patients remained transparent several decades and have to be extracted late in the life! From the data included in the manuscript it does not appear that the authors were able to look at the MS/MS data to unequivocally prove that normal and cataract lenses had so many mutations, summarized in Table S3 and S4.
A: We corrected the text to explain more clearly on the mutations. We agree with the reviewer that all of the mutations reported in our MS-based analysis are not real. They were possible mutations and might include false positives due to deficiency in MS data and analysis algorithm. Thus, we did not focus on all mutations detected in all samples. We eliminated the false positives included in detected mutations by only focusing on cataract-specific mutations. We detected 112 possible mutations in our analysis. If the 112 mutations were random, the mutations should have been uniformly distributed across seven samples. However, we observed that twelve mutations in Table 1 were observed in both cataract samples but never in other normal samples. We emphasize that such occurrence can hardly happen by chance. The probability of randomly observing at least 12 mutations only in those same two samples by chance is 9.264×10^-10. This calculation is described in page 12. To avoid any misunderstanding, we also revised the phrase ‘novel mutations’ as ‘possible mutations’ in Table S4.
Additionally, it is very surprising that the investigators who originally analyzed those lens proteins and prepared data sets did not recognize such large number of mutations.
A: The original works focused on modifications such as deamidation in aged lens. To find mutations using MS data is not trivial since the MS-based mutation analysis is computationally very expensive. MODplus, recently developed, enables the comprehensive detection of mutations in this study.
Results:
Lines 114 and 233: statements regarding the secondary/tertiary structural changes. The authors should perform CD spectroscopy of purified WT and double mutants and analyze α-helical, β-sheet and random structure content and include the results in the Suppl section.
A: We corrected the expression to “conformational state”
Fig 2 a and 4 C: It will be good to estimate the ∆G of the protein unfolding and include in the results.
A: That's a very good suggestion, but we cannot estimate free energy changes.
Fig 5 b: It is not clear whether the unfolding is complete at 60 mins. Data from an additional time point should be included to confirm this.
A: It showed a significant difference from 10 minutes, so the difference between WT and mutant was confirmed as the experimental result up to 60 minutes, because it is fully saturated.
Oxidation of gamma D-mutants: From the data shown in Fig 6 and Table 2 it is clear that the double mutant is more susceptible to oxidation. It is also known that gamma crystallins are more susceptible to oxidation and oxidation can result in protein conformation changes, including the exposure of hydrophobic regions. Data presented in Fig 4 shows that the double mutant displays higher degree of conformation change and decreased stability. On examining the methods section it appears that the double mutant (P24S/S31G) was not kept in buffers containing reducing agent to prevent oxidation of Cys in gamma protein. Therefore, it is possible that the proteins that showed increased ThT binding and ANS binding were oxidized and the conformation change observed was due to protein oxidation rather than mutation by itself. This should be clarified.
A: We agree with reviewer’s comment. It is known that crystallin mutations increase susceptibility to oxidation and induce conformation changes which decrease protein stability. Although the causal relationship between them is difficult to confirm, oxidation susceptibility changes in mutants are occurred from conformational changes.
Solubility of mutants: The authors have not performed solubility tests for the proteins or not shown any data that demonstrates aggregation and precipitation of mutants. It will be good to provide this information in the manuscript. If the proteins have decreased solubility, how they were purified/recovered.
A: As suggested, we performed two additional experiments to examine the aggregation of two mutants. We found that the mutants are more readily aggregated than the WT in vitro and in vivo. We added the Figure 2c and 4d, showing the turbidity increases of mutant proteins, and Fig. S5, observing the protein aggregation increase in the cells transfected with mutants.
Although the data were not presented, the mutant was less expressed in the cell transfected with same amount of DNA compared to WT, possibly because the mutant had low solubility.
Methods:
Lines 370 to 381- The write up is similar to the method section in ref 24. This section should be presented between quotes with proper acknowledgement. Otherwise, the authors need to re-write this part of the manuscript.
A: As you suggested, we added citation in methods.
Suppl section:
Figure S4: Realign “ – “ and “+” H2O2” above the gel. What are those bands above dimers in samples without ME ? The so called monomers in lanes showing samples without ME are seen as double bands. Why ?
A: As you suggested, we realigned the display of the Figure S4. The bands above the dimer disappear in the presence of β-ME, so it is assumed that they are oligomers of βB2-crystallin. Also, the reason that the monomer form is two bands is that there is an intra-disulfide bond in the beta crystallin population.
Figure S6 legend- correct “spetra” as “spectra”
A: We corrected to “spectra”
Figure S8: What is the source of the data? If this is a replot, based on literature data appropriate reference and permission to reproduce has to be stated.
A: We corrected the text to clarify what we have done. The fractions of crystallin proteins were calculated from MODplus search in this work. We revised the sentence in line 402 in page 12 as follows: “The fractions of crystallin proteins in each lens sample identified by MODplus are also shown in Fig S8.”
Table S4, a and b: The authors claim that they have identified 37 mutations in 70 and 93 yr old cataract lenses. It is very surprising that in spite of so many mutations those lenses remained transparent (?) for 7 to 9 decades of life since the authors also show that the double mutants of beta B2 and gamma D are unstable, aggregate and likely cataractogenic.
A: That's an interesting question. This might be the difference in the degree of aggregation. Crystallin mutants certainly have a tendency to agglomerate, but they appear to be finely agglomerated rather than completely agglomerated agglomerates, so it is thought that they may spread without sinking into the solution.
We hope that based on our responses, you will find our revised manuscript acceptable for publication in INTERNATIONAL JOURNAL OF MOLECULAR SCIENCES. We will be pleased to address any additional comments or questions the reviewers might have. We look forward to hearing from you soon.
Sincerely Yours,
Kong-Joo Lee
Professor
Graduate School of Pharmaceutical Sciences
College of Pharmacy
Ewha Womans University
Seoul, Korea 120-750
TEL#: 82-2-3277-3038
e-mail: [email protected]

Round 2
Reviewer 2 Report
In few places spelling of "crystalline" should be corrected as "crystallin"